# Spectraformer: A Unified Random Feature Framework for Transformer

## Abstract

Linearization of attention using various kernel approximation and kernel learning techniques has shown promise. Past methods use a subset of combinations of component functions and weight matrices within the random features paradigm. We identify the need for a systematic comparison of different combinations of weight matrices and component functions for attention learning in Transformer. In this work, we introduce *Spectraformer*, a unified framework for approximating and learning the kernel function in linearized attention of the Transformer. We experiment with broad classes of component functions and weight matrices for three textual tasks in the LRA benchmark. Our empirical findings indicate that different kernels are good at different tasks and that kernel choice is fundamental to performant models. Our code is available at: `https://anonymous.4open.science/r/spectraformer-8A97`.

## 1 Introduction

Transformer (Vaswani et al., 2017) has revolutionized the landscape of natural language processing (NLP) and forms the basis of almost all state-of-the-art language models. Its influence has reached beyond NLP into computer vision (Han et al., 2023), speech processing (Lin et al., 2022) and other fields. Compared to its precursors like the LSTMs, Transformer leverages parallelism since it is fully based on the attention mechanism. Therefore, the performance of Transformer is tied to the effective use of attention. Attention in the Original Transformer (referred to as '**OT**' hereafter) uses the softmax function which is quadratic in time complexity. However, softmax is not fundamental to attention. The softmax serves the role of a **compatibility function**, which captures the degree of compatibility between two tokens (see Equations 1 and 2). Extensive experiments with other compatibility functions show that the softmax is not the only possibility, and that the 'best' **compatibility function** depends on the task (Tsai et al., 2019).

Replacing the softmax formulation with alternatives may allow us to reduce the time complexity of attention computation. In this paper, we focus on the subset of alternatives that employs kernel functions. Kernel functions have been historically used in machine learning algorithms to simplify the estimation of parameters (Hofmann et al., 2008). Tsai et al. (2019) show that attention can be formulated using kernels and then be linearized (Katharopoulos et al., 2020), given the feature map of the respective attention kernel. Random feature-based algorithms (Liu et al., 2022) are a family of algorithms, inspired by spectral analysis, which provides the feature map associated with a kernel. A random feature consists of three components: component function, weight matrix, and component weight (see Section 3). Due to their robustness and flexibility, random features give rise to '*random feature Transformers*', the family of Transformers with linearized attention via random features (hereby abbreviated as RF), first with the work by Choromanski et al. (2021). This is seen to lead to three strands of research: (A) **Optimization of weight matrices** either by making the matrices have lower time or space complexity, or by incorporating beneficial properties like orthogonality to reduce variance (Yu et al., 2016; Choromanski et al., 2021; 2022; Reid et al., 2023); (B) **Enhancement of component functions** by either: (i) finding a function with a tighter variance or approximation bound, or (ii) be engineered to output with numerical stability and be bounded (Choromanski et al., 2021; Likhosherstov et al., 2022; 2023); (C) Parameterization of weight matrices to perform kernel learning instead of kernel approximation (Chowdhury et al., 2022). (A) and (B) seek to improve approximation quality via approximation error and variance reduction. (C) dispenses with the approximation of the attention kernel (typically the softmax), and makes the kernel itself learnable, to attain a better

performance. However, the three have been explored separately, creating a situation where there are different ways to construct the random features. Additionally, past works only explore certain combinations of component functions and weight matrices, creating disjoint overlaps and gaps in the literature. Finally, the weight matrices are often those explicitly designed in the context of kernelized attention in the Transformer (Yu et al., 2016; Choromanski et al., 2021; 2022; Reid et al., 2023), ignoring those which are popular in the kernel methods literature (e.g., Liu et al. (2022)). These limitations necessitate a unified framework that allows the comparison of different combinations in search of the best-performing one.

We introduce *Spectraformer*, a unified framework of random feature-based attention in Transformer. It utilizes **spectral** analysis-inspired random features to construct linearized attention in **Transformers**. *Spectraformer* allows for the experimentation of various combinations of weight matrices and component functions. We will open-source our codebase upon acceptance. Through our benchmarking experiments on **three textual Long Range Arena (Tay et al., 2021) tasks** and **18 combinations** of weight matrices and component functions, *Spectraformer* enables novel combinations of component functions and weight matrices from the kernel literature to be experimented easily in the Transformer setting. This process makes possible the discovery of combinations that outperform existing random feature-based Transformers in terms of performance, training time, and peak memory consumption. Various novel Spectraformer combinations are more accurate than the previous SOTA models, SADERF-ORF (a.k.a., FAVOR# (Likhosherstov et al., 2023)), OPRF-ORF (a.k.a., FAVOR++ (Likhosherstov et al., 2022)), PosRF-ORF (a.k.a., FAVOR+ (Choromanski et al., 2021)), whilst significantly reducing training time and memory consumption over both. We hope that *Spectraformer* will enable the mechanism to benchmark more random feature-based Transformers for these and other tasks. Our contributions are:

1. Our novel framework, *Spectraformer*, which generalizes over all past works in linearized attention with kernel approximation and kernel learning.

2. We demonstrate how Spectraformer can discover novel SOTA models in the random feature Transformer family for the LRA benchmark and that these models are more performant than existing random feature Transformers.

3. We empirically identify the relationship between performant combinations and task characteristics, aiding the right kernel choice when tackling a task.

4. Our code is currently available for review, but they will be publicly released upon acceptance. We detail guide on how to incorporate future kernel works into Spectraformer. With this work, every future kernel work that is introduced can easily be incorporated into the transformer.

## 2 PRELIMINARIES

We first cover the theoretical foundations to this work. A list of commonly used acronyms is available in Table 1.

Table 1: Table of acronyms

| Acronym | Description |
|---------|-------------|
| OT (Vaswani et al., 2017) | Original Transformer |
| RF (Rahimi & Recht, 2007) | Random Feature method |
| RBF | Radial Basis Function kernel (also called the Gaussian kernel) |
| TrigRF (Rahimi & Recht, 2007) | Trigonometric RF (component function) |
| PosRF (Choromanski et al., 2021) | Positive RF (component function) |
| OPRF (Likhosherstov et al., 2022) | Optimized positive RF (component function) |
| SADERF (Likhosherstov et al., 2023) | Simplified Asymmetric Dense-Exponential RF (component function) |
| ORF (Yu et al., 2016) | Orthogonal RF (weight matrix) |
| SORF (Yu et al., 2016) | Structural ORF (weight matrix) |
| QMC (Avron et al., 2016) | Quasi- Monte Carlo (weight matrix) |
| MM (Shen et al., 2017) (Liu et al., 2022) | Moment Matching (weight matrix) |
| SGQ (Dao et al., 2017) | Sparse Grid Quadrature (weight matrix) |
| FastFoodL(Yang et al., 2015) | FastFood Learnable (weight matrix) |

## 2.1 Kernelized Attention

OT attention (Vaswani et al., 2017) can be defined using the unified attention model (Galassi et al., 2021) as follows. Given a collection of inputs and a learning objective, **attention** mechanism learns the attention matrix $\mathbf{A}$ which captures the associative information between pairs of tokens in a sequence of length $N$, with each row $\mathbf{A}_i$ being the representation of a token $i$ with every other token $j \in [1, N]$. This is done by creating three learnable representations of each token $i$ called **query** ($\boldsymbol{q}_i$), **key** ($\boldsymbol{k}_i$), and **value** ($\boldsymbol{v}_i$). The attention representation $\mathbf{A}_i$ is calculated as a weighted sum of value $\boldsymbol{v}_j$ given **attention weights** $a_j$. The general equation for $\mathbf{A}_i$ is given as:

$$\boldsymbol{A}_i = \sum_{j=1}^N \frac{f(\boldsymbol{q}_i, \boldsymbol{k}_j)}{\sum_l f(\boldsymbol{q}_i, \boldsymbol{k}_l)} \boldsymbol{v}_j = \sum_{j=1}^N a_j \boldsymbol{v}_j; \quad a = g(e); \quad e = f(\boldsymbol{q}_i, \boldsymbol{k}_j) \tag{1}$$

Attention weights are generated using a **distribution function** $g$ applied on an energy score $e$. The energy score is captured by applying a **compatibility function** on a query $\boldsymbol{q}_i$ and a set of keys $\mathbf{K}$, in order to compute a score of a specific relationship between the pairs. The attention in the OT is called the 'scaled dot-product' which defines the compatibility function $f$ and the distribution function $g$ as follows:

$$f(\boldsymbol{q}_i, \mathbf{K}) = \frac{\boldsymbol{q}_i^T \mathbf{K}}{\sqrt{d_k}} \quad g(\boldsymbol{x}_j) = \frac{\exp(\boldsymbol{x}_j)}{\sum_l \exp(\boldsymbol{x}_l)} \tag{2}$$

We can then have the attention matrix (with $\exp(.)$ being applied element-wise) as:

$$\boldsymbol{A}_i = \sum_{j=1}^N \frac{\exp(\boldsymbol{q}_i \boldsymbol{k}_j^T / \sqrt{d_k})}{\sum_l \exp(\boldsymbol{q}_i \boldsymbol{k}_l^T / \sqrt{d_k})} \boldsymbol{v}_j \quad A = softmax(\frac{\boldsymbol{Q}\boldsymbol{K}^T}{\sqrt{d_k}})\boldsymbol{V} \tag{3}$$

with $\boldsymbol{Q}, \boldsymbol{K}, \boldsymbol{V}$ being the matrix of queries, keys, and values respectively with each row being $\boldsymbol{q}_i, \boldsymbol{k}_i, \boldsymbol{v}_i$.

We now discuss the intrinsic connection between kernel and attention. With the compatibility function $f$ as a kernel $f(\boldsymbol{q}_i, \boldsymbol{k}_j) = \mathcal{K}(\boldsymbol{q}_i, \boldsymbol{k}_j)$, Equation 1 can be rewritten as:

$$\boldsymbol{A}_i = \sum_{j=1}^N \frac{\mathcal{K}(\boldsymbol{q}_i, \boldsymbol{k}_j)}{\sum_l \mathcal{K}(\boldsymbol{q}_i, \boldsymbol{k}_l)} \boldsymbol{v}_j \tag{4}$$

As shown in Tsai et al. (2019), this is indeed the Nadaraya-Watson Kernel Estimator, with $\mathbb{E}_{p(\boldsymbol{k}_j | \boldsymbol{q}_i)}[\boldsymbol{v}_j | X = \boldsymbol{k}_j] = \boldsymbol{A}_i, l_i(\boldsymbol{k}_j) = p(\boldsymbol{k}_j | \boldsymbol{q}_i), Y_i = \boldsymbol{v}_j$:

$$\mathbb{E}_{l_i}[Y_i | X = \boldsymbol{u}] = \sum_{i=1}^N l_i(\boldsymbol{u}) Y_i; \quad l_i(\boldsymbol{u}) = \frac{\mathcal{K}(\frac{\boldsymbol{u} - \boldsymbol{x}_i}{h})}{\sum_{j=1}^n \mathcal{K}(\frac{\boldsymbol{u} - \boldsymbol{x}_j}{h})} \tag{5}$$

Specifically, the scaled dot-product attention in the OT in Equation 3 corresponds to the softmax kernel (see Equation 6).

$$\mathcal{K}_{softmax}(\boldsymbol{q}_i, \boldsymbol{k}_j) = \exp(\boldsymbol{q}_i^T \boldsymbol{k}_j / \sqrt{d_k}) \tag{6}$$

## 2.2 Linearizing Kernelized Attention via Random Features

Since a kernel is the inner product of two vectors in some space with respect to some feature map $\phi$, i.e., $\mathcal{K}(\boldsymbol{x}, \boldsymbol{y}) = \phi(\boldsymbol{x})\phi(\boldsymbol{y})^T$, then we can rewrite Equation 4 as:

$$\boldsymbol{A}_i = \frac{\sum_j \phi(\boldsymbol{q}_i)\phi(\boldsymbol{k}_j)^T \boldsymbol{v}_j}{\sum_l \phi(\boldsymbol{q}_i)\phi(\boldsymbol{k}_l)^T} = \frac{\phi(\boldsymbol{q}_i) \sum_j \phi(\boldsymbol{k}_j) \otimes \boldsymbol{v}_j}{\phi(\boldsymbol{q}_i) \sum_l \phi(\boldsymbol{k}_l)^T} \tag{7}$$

If we pre-compute $\sum_j \phi(\boldsymbol{k}_j) \otimes \boldsymbol{v}_j$ and $\sum_l \phi(\boldsymbol{k}_l)^T$, the entire term becomes $O(1)$ and we only need to compute Equation 7 $N$ times for each $i$, resulting in an attention matrix calculated in linear time $O(N)$. We now refer to Equation 7 as **linearized attention** (Katharopoulos et al., 2020).

All that is left is to retrieve the feature map $\phi$, the process of which is called **kernel approximation** (see Section 2.3), which corresponds to the kernels we specify. There are many popular kernels

from which we can choose, but typically, we want to approximate the softmax kernel in Equation 6. However, since the softmax relates to the $\mathcal{K}_{RBF}(\boldsymbol{x}, \boldsymbol{y}) = \exp(-\frac{||\boldsymbol{x}-\boldsymbol{y}||^2}{2})$ via the following equation:

$$\mathcal{K}_{softmax}(\boldsymbol{x}, \boldsymbol{y}) = \exp(\frac{||\boldsymbol{x}||^2}{2})\mathcal{K}_{RBF}(\boldsymbol{x}, \boldsymbol{y})\exp(\frac{||\boldsymbol{y}||^2}{2}) \tag{8}$$

Approximating the RBF is easier than the softmax directly in the method we introduce in Section 2.3, hence we approximate the softmax indirectly via the RBF in Equation 8.

Oftentimes, however, we might not want to specify a kernel, since kernel choice is indeed a hyperparameter and the softmax is by no means essential or necessary as shown in Tsai et al. (2019). Then kernel learning would be a more suitable option, the detail of which is discussed in Section 2.4. Notwithstanding, we first introduce the main kernel approximation technique: random features.

## 2.3 RANDOM FEATURES FOR KERNEL APPROXIMATION

Of the kernel approximation family of methods, the most successful and appropriate for linearized attention in Transformer has been the random features approach (Rahimi & Recht, 2007). Estimating kernels using random features relies on a fundamental insight from harmonic analysis called Bochner's theorem (Rudin, 1990) stated as follows:

*A continuous shift-invariant kernel $\mathcal{K}(\boldsymbol{x}, \boldsymbol{y}) = \mathcal{K}(\boldsymbol{x} - \boldsymbol{y})$ on $\mathbb{R}^d$ is positive definite if and only if $\mathcal{K}(\boldsymbol{\delta})$ is the Fourier transform of a non-negative measure $p(\boldsymbol{\omega})$. If $\mathcal{K}(\boldsymbol{\delta})$ is properly scaled, that is $\mathcal{K}(0) = 1$ then the measure $p(\boldsymbol{\omega})$ is a proper probability distribution.*

$$\mathcal{K}(\boldsymbol{\delta}) = \int p(\boldsymbol{\omega}) \exp(i\boldsymbol{\omega}\boldsymbol{\delta})\mathrm{d}\boldsymbol{\omega} \tag{9}$$

When approximating the integration in this equation via Monte Carlo sampling with $s$ samples, $\boldsymbol{\omega} \sim p(\boldsymbol{\omega})$, we can obtain the feature map:

$$\begin{aligned} \mathcal{K}(\boldsymbol{\delta}) \quad &= \mathbb{E}[\phi_{\boldsymbol{\omega}}(\boldsymbol{x})^T \phi_{\boldsymbol{\omega}}(\boldsymbol{y})] \approx <\phi_{\boldsymbol{\omega}}(\boldsymbol{x}), \phi_{\boldsymbol{\omega}}(\boldsymbol{y})> \\ \phi_{\boldsymbol{\omega}}(\boldsymbol{x}) \quad &= \frac{1}{\sqrt{s}}\big[\exp(-\mathrm{i}\boldsymbol{\omega}_1^\top \boldsymbol{x}), \dots, \exp(-\mathrm{i}\boldsymbol{\omega}_s^\top \boldsymbol{x})\big]^\top \end{aligned} \tag{10}$$

## 2.4 RANDOM FEATURES FOR KERNEL LEARNING

Previous random feature techniques are robust in approximating kernels for specific learning problems. However, kernels may be chosen using heuristics and by convention from a popular subset. The kernel is very much a hyperparameter. This could pose a problem since there is no free lunch in learning as much as in kernel choice (Schölkopf & Smola, 2018). Tsai et al. (2019) has shown experimental results that this is also true in the context of attention in Transformer. Hence, instead of picking a kernel for the attention, we could learn the kernel via **kernel learning** an established kernel methods technique. Kernel learning has shown to be effective in general (Tompkins et al., 2019; Wilson & Adams, 2013; Oliva et al., 2016) and in the case of attention via Gaussian Mixture Model (GMM), learnable weight matrices (FastFood), deep generative model (DGM) (Chowdhury et al., 2022). Kernel learning is achieved by making the weight matrix $\boldsymbol{W}$ learnable where $\boldsymbol{\omega}_i$ is the output of a function parameterizing $p(\boldsymbol{\omega})$.

The kernelized attention via random features discussed here is a popular approach, however it is by no means the only one. A more detailed discussion is offered in related work in Section A.2. Specifically, Section A.2.1 provides alternative kernelized attention formulations and Section A.2.2 provides alternative kernel approximations to random features.

## 3 SPECTRAFORMER

*Spectraformer* is a unified random feature framework that allows for the combination of any weight matrix with any component function to be used as kernelized attention in Transformer. The general equation for random feature is shown in Equation 11, generalized from Equation 10, based on Liu et al. (2022); Choromanski et al. (2021).

$$\phi_{\boldsymbol{\omega}}(\boldsymbol{x}) = \quad \frac{1}{\sqrt{s}}\big[a_1 f_1(\boldsymbol{\omega}_1, \boldsymbol{x}), ..., a_s f_1(\boldsymbol{\omega}_s, \boldsymbol{x}), ...,$$
$$a_1 f_l(\boldsymbol{\omega}_1, \boldsymbol{x}), ..., a_s f_l(\boldsymbol{\omega}_s, \boldsymbol{x})\big]^\top \tag{11}$$
$$\boldsymbol{W} = \quad [\boldsymbol{\omega}_1, ..., \boldsymbol{\omega}_s]^T \in \mathbb{R}^{s*d}$$

*Spectraformer* is shown in Figure 1. The attention of the OT (on the left-hand side) is replaced by the attention of the Spectraformer which pre-computes the Hadamard (element-wise) product of values $\boldsymbol{v}$ and transformed keys $\phi(\boldsymbol{k})$ scaled by $\phi(\boldsymbol{k})$, the product of which is multiplied with the transformed queries $\phi(\boldsymbol{q})$. The linear map $\phi$ is composed from weight matrices and component functions. The linear map has $s$ components with $a$ being the component weight (see Equation 11). The component weight $a$ can either be fixed (typically at $a = 1$) or learnable as proposed by various kernel works (Liu et al., 2022). Due to the fact that the majority of random feature Transformers (e.g., Likhosherstov et al. (2022)) have $a = 1$, we follow this setting for consistent comparison between combinations.

The figure and the equation together highlight the three strands of research as mentioned previously in Section 1: (A) Weight matrix approximator: constructing $\boldsymbol{W}$ more effectively; (B) Weight matrix learner: parameterizing $\boldsymbol{W}$ instead of sampling; (C) Component function: constructing $f_j$ more effectively.

## 3.1 WEIGHT MATRICES

In Spectraformer, the weight matrix $\boldsymbol{W}$ which is either an approximator (from the families of (e.g., ORF and SORF (Yu et al., 2016)), Unit-cube (e.g., QMC (Avron et al., 2016), MM (Shen et al., 2017)), Quadrature (e.g., SGQ (Dao et al., 2017)). or a learner (FastFood$_L$ (Chowdhury et al., 2022)). We term the matrix $\boldsymbol{W}$ as weight matrix instead of random matrix since although $\boldsymbol{W}$ acts like a

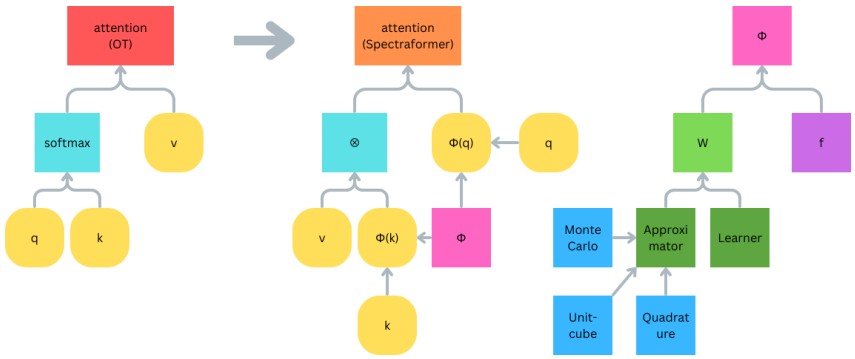

Figure 1: Spectraformer framework based on Equation 11

random matrix ($\boldsymbol{\omega}_i \sim p(.)$, see Equation 10), it is not guaranteed to be one. In the case of weight matrix approximator, they approximate a random matrix and in the case of weight matrix learner, they parameterize a weight matrix, thereby implicitly learning a distribution $p(.)$ associated with such $\boldsymbol{W}$. The weight matrix is given in Definition 3.1.

**Definition 3.1.** Provided that a matrix can substitute for a random matrix in Equation 11, and the solution for this equation given such matrix and $f$ as TrigRF being the solution for Equation 9, i.e., as an unbiased estimator of $\mathcal{K}$, and TrigRF being $f_1 = f_l = \exp(-i\boldsymbol{\omega}^T \boldsymbol{x})$ (see Equation 10) then such a matrix is a weight matrix.

It follows from Definition 3.1 that for any weight matrix $\boldsymbol{W}$, $f = TrigRF$ is a valid component function. Spectraformer **enables weight matrix approximation** in terms of three families based on how they solve the intractable integral in Equation 9:

- **Monte Carlo sampling**-based weight matrix involves approximating the integral of Equation 9 using Monte Carlo sampling with the solution given by Equation 10 (Rahimi & Recht, 2007) with $\boldsymbol{W}$ being the 'Base' random matrix of $p(.)$, $\boldsymbol{\omega}_i \sim p(.)$. The random matrix $\boldsymbol{W}$ can be constructed more efficiently (either reducing time or space complexity) with FastFood (Le et al., 2013). The random matrix $\boldsymbol{W}$ can also be enforced to form geometrical couplings such as orthogonality to

reduce approximation error with ORF, SORF (Yu et al., 2016). Both these methods produce valid random matrices.

- **Unit-cube sampling**-based weight matrix transforms Equation 9 to an integral on the unit cube $[0, 1]^d$, then performs the approximation with uniform and independent points, thus reducing variance. The first approach is QMC (Avron et al., 2016), where $\boldsymbol{W} : \boldsymbol{\omega}_i = \Phi^{-1}(\boldsymbol{t}_i)$, $\Phi$ being the cumulative distribution function (CDF) associated with $p(.)$ and $\boldsymbol{t}_i$ being a low discrepancy sequence. $\Phi^{-1}(\boldsymbol{t}_i) \sim p(.)$ whilst having a lower variance than direct sampling from $p(.)$. MM (Shen et al., 2017; Liu et al., 2022) improves over QMC by replacing $\Phi^{-1}$ with a moment matching scheme $\widetilde{\Phi}^{-1}$ on $\Phi^{-1}$.

- **Quadrature**-based weight matrix uses quadrature rules with non-uniform deterministic weights. The main approach explored is SGQ (Dao et al., 2017) which uses one-dimensional Gaussian quadrature rule by assuming $\mathcal{K}$ factorizes with respect to the dimensions. Smolyak rule is further implemented to alleviate the curse of dimensionality.

Liu et al. (2022) show that in various kernel benchmark datasets (among combination with TrigRF), QMC and ORF have consistently high performance. Due to the significant number of weight matrices in random features literature, we decide to cover only the most prominent ones. Appendix A.3 provides the technical details of the approximators discussed here and those that we left out.

Spectraformer also **allows to learn weight matrices** by learning $p(.)$ via parameterizing $\boldsymbol{W}$. Yang et al. (2015) introduce several of these parametrization schemes, which are then adapted into the attention setting by Chowdhury et al. (2022). The best performing method is via making the constituent matrices of FastFood learnable, which we term FastFood$_L$. In general, a weight matrix learner is valid as long as it can implicitly model any distribution. In addition the TrigRF, PosRF has also been applied on FastFood$_L$. Technical details and other learners we have not covered are explored in Appendix A.4. We now explore existing works which have developed from Rahimi & Recht (2007) (see Equation 9, 10) and how they lead to the development of Spectraformer.

## 3.2 COMPONENT FUNCTIONS

Spectraformer incorporates component functions $f$ such as PosRF, OPRF, SADERF. Component functions in Spectraformer combine each weight matrix row $\boldsymbol{\omega}_i$ with the input $\boldsymbol{x}$. The base case of the component function is TrigRF. Component function is given in Definition 3.2

**Definition 3.2.** A function of $\boldsymbol{x}$ with respect to $\boldsymbol{\omega}$ is a valid component function if and only if $\mathcal{K} \approx \phi_{\boldsymbol{\omega}}(\boldsymbol{x})^T \phi_{\boldsymbol{\omega}}(\boldsymbol{y})$ given that $\boldsymbol{\omega}_i \sim p(.)$ in Equation 10.

It follows that all component functions have provided theoretical guarantees given that $\boldsymbol{W}$ is a random matrix. In addition, PosRF, OPRF, and SADERF have also been proven theoretically to approximate $\mathcal{K}_{RBF}$ given $\boldsymbol{W}$ being the ORF.

While **TrigRF**(Rahimi & Recht, 2007) is the base component function and guarantees $\mathcal{K}(\boldsymbol{\delta})$ to be close to $\phi_{TrigRF}(\boldsymbol{x}) \cdot \phi_{TrigRF}(\boldsymbol{y})$ due to uniform convergence, it is typically evaluated using only the real component of $\exp(-i\boldsymbol{\omega}_j^T \boldsymbol{x})$. This leads to $\cos$ and $\sin$ functions which can produce unstable behavior, especially when $K(\boldsymbol{\delta})$ is close to 0, where variance is infinite (Choromanski et al., 2021; Likhosherstov et al., 2022). Therefore, TrigRF is avoided in Spectraformer.

- **PosRF** (Choromanski et al., 2021) is introduced in Spectraformer as a simple modification over TrigRF to counter this instability. PosRF guarantees positive function output and its variance approaching 0 as $K(\boldsymbol{\delta})$ approaches 0. Likhosherstov et al. (2022) generalize TrigRF and PosRF to GERF.

- **OPRF** (Likhosherstov et al., 2022) is one of the solution to the reduction of the variance equation of GERF. OPRF is both positive and bounded (whereas PosRF is positive and unbounded). Likhosherstov et al. (2023) generalize GERF to dense matrix instead of scalar parameters leading to DERF.

- **SADERF** (Likhosherstov et al., 2023) is one of the solutions to the reduction of the variance equation of DERF. Whilst DERF has many solutions, they all rely on operations which are incompatible with many deep learning libraries on GPU and TPU like SVD and eigen decompositions. SADERF is an extension of GERF with a tighter variance bound on DERF.

The alternatives for component functions allow Spectraformer to approximate the RBF with $p(.)$ corresponding to the Gaussian. Likhosherstov et al. (2023) and Likhosherstov et al. (2022) show that SADERF-ORF is the SOTA combination followed by OPRF-ORF and PosRF-ORF. Technical details and other component function(s) not covered are explored in Appendix A.5.

Table 2: Summary of research gap in kernelized attention. Rows are the component functions, columns are the weight matrices. For each cell, the top item is the cited work for this combination for typical kernel tasks, while the bottom item is the cited work for Transformer-based tasks. $\times$ indicates no prior work for the combination. Spectraformer explores all combinations listed here, excluding 'Base' due to exhaustive work as seen in the table.

| | | TrigRF | PosRF | OPRF | SADERF |
|---|---|---|---|---|---|
| Monte Carlo | Base | (Rahimi & Recht, 2007)
(Choromanski et al., 2021) | (Choromanski et al., 2021)
(Choromanski et al., 2021) | (Likhosherstov et al., 2022)
(Likhosherstov et al., 2022) | (Likhosherstov et al., 2023)
(Likhosherstov et al., 2023) |
| | ORF | (Yu et al., 2016)
(Choromanski et al., 2021) | (Choromanski et al., 2021)
(Choromanski et al., 2021) | (Likhosherstov et al., 2022)
(Likhosherstov et al., 2022) | (Likhosherstov et al., 2023)
(Likhosherstov et al., 2023) |
| | SORF | (Yu et al., 2016)
$\times$ | $\times$
$\times$ | $\times$
$\times$ | $\times$
$\times$ |
| Unit-cube | QMC | (Avron et al., 2016)
$\times$ | $\times$
$\times$ | $\times$
$\times$ | $\times$
$\times$ |
| | MM | (Shen et al., 2017)
$\times$ | $\times$
$\times$ | $\times$
$\times$ | $\times$
$\times$ |
| Quadrature | SGQ | (Dao et al., 2017)
$\times$ | $\times$
$\times$ | $\times$
$\times$ | $\times$
$\times$ |
| Learner | FastFood$_L$ | (Yang et al., 2015)
(Chowdhury et al., 2022) | $\times$
(Chowdhury et al., 2022) | $\times$
$\times$ | $\times$
$\times$ |

## 3.3 UTILITY

There has been an unsystematic attempt at combining component functions and weight matrices. On one hand, most component functions have had theoretical and experimental results in combining with Base and ORF. On the other hand, most weight matrices have had results in combining with TrigRF or PosRF. This means that a lot of potential combinations have not been studied previously. Table 2 shows the wide research gap across 14 combinations which have never been studied in either the kernel or Transformer setting. Spectraformer highlights this research gap. Excluding TrigRF due to its instability and Base due to its under-performance, we conduct experiments on 18 combinations.

*Spectraformer* is a generic random feature framework that allows for the combination of any weight matrix with any component function provided that the weight matrix satisfies Definition 3.1 and the component function satisfies Definition 3.2. We note that *Spectraformer* is an experimental rather than a theoretical framework. Due to the large number of weight matrices and component functions, it is infeasible for the time being to derive a general mathematical proof for the bounds and convergence of *Spectraformer*. Therefore, we seek to validate its feasibility and show its ability to discover novel and performant combination via experimental results.

## 3.4 ADDING NEW CODE TO SPECTRAFORMER

We now discuss how to add a new component function or weight matrix to the base code for further implementation. New component functions need to satisfy Definition 3.2 and new weight matrices need to satisfy Definition 3.1.

### 3.4.1 COMPONENT FUNCTION

1. Add the new component function f to `src/models/component_functions.py`, the arguments should include data (the input), and other optional parameters.

2. Import f and add a new entry to `FastAttention.comp_functions[f_name] = f` in `src/models/attention_performer.py` (line 176)

### 3.4.2 WEIGHT MATRIX

1. Add the new weight matrix w to `src/models/weight_matrix_approx.py` or `src/models/weight_matrix_learner.py`, the arguments should include `nb_rows` (number of rows), `nb_cols` (number of columns) and `device`.

2. Import  w  and  add  the  if  clause  and  $w$  function  call  in `src/models/attention_performer.py` (line 208)

## 4 RESULTS

### 4.1 EXPERIMENT DETAILS

Table 3: Experimental results of Spectraformer variants on the LRA benchmark, over five seeds. We report mean accuracy (on the test set), mean time (training time (hour)), and mean memory (peak memory consumption (GB)). $L$: ListOps, $T$: Text, $R$: Retrieval, and $\mu$ for the average of the three. Entries are ranked by $\mu$ accuracy. The previous RF SOTAs (SADERF-ORF (Likhosherstov et al., 2023) and OPRF-ORF (Likhosherstov et al., 2022)) are in bold. We note that via Spectraformer, we find various variants outperforming these previous RF SOTAs in terms of accuracy, with different combinations offering different trade-off between accuracy, time, and memory. Entries are sorted by mean accuracy.

| | Accuracy (%) ↑ | | | | Time (hour) ↓ | | | | Memory (GB) ↓ | | | |
|---|---|---|---|---|---|---|---|---|---|---|---|---|
| | L | T | R | $\mu$ | L | T | R | $\mu$ | L | T | R | $\mu$ |
| OPRF-FastFoodL | 37.55 (0.48) | 64.41 (0.62) | 77.70 (0.33) | 59.89 | 1.07 | 2.07 | 2.12 | 1.75 | 0.86 | 1.72 | 1.68 | 1.42 |
| OPRF-MM | 38.08 (0.53) | 60.40 (0.85) | 81.09 (0.18) | 59.86 | 0.68 | 1.26 | 1.24 | 1.06 | 1.36 | 2.71 | 2.56 | 2.21 |
| PosRF-MM | 37.06 (0.37) | 61.87 (1.79) | 80.58 (0.53) | 59.84 | 0.56 | 1.05 | 1.06 | 0.89 | 1.17 | 2.31 | 2.10 | 1.86 |
| **OPRF-ORF** | **38.34 (0.22)** | **60.16 (0.79)** | **80.88 (0.17)** | **59.80** | **0.68** | **1.26** | **1.25** | **1.06** | **1.36** | **2.71** | **2.56** | **2.21** |
| SADERF-QMC | 37.37 (0.38) | 61.14 (1.48) | 80.84 (0.14) | 59.78 | 0.68 | 1.25 | 1.29 | 1.07 | 1.44 | 2.86 | 2.69 | 2.33 |
| PosRF-QMC | 37.11 (0.09) | 61.69 (0.96) | 80.55 (0.13) | 59.78 | 0.56 | 1.05 | 1.05 | 0.89 | 1.17 | 2.31 | 2.10 | 1.86 |
| SADERF-MM | 37.10 (0.22) | 60.68 (1.88) | 81.13 (0.17) | 59.64 | 0.68 | 1.25 | 1.29 | 1.07 | 1.44 | 2.86 | 2.69 | 2.33 |
| **SADERF-ORF** | **37.10 (0.19)** | **60.39 (2.08)** | **81.05 (0.22)** | **59.51** | **0.68** | **1.25** | **1.28** | **1.07** | **1.44** | **2.86** | **2.69** | **2.33** |
| OPRF-QMC | 37.69 (0.62) | 59.94 (0.59) | 80.38 (0.49) | 59.34 | 0.68 | 1.26 | 1.26 | 1.07 | 1.36 | 2.71 | 2.56 | 2.21 |
| SADERF-SGQ | 37.11 (0.21) | 62.46 (0.54) | 78.38 (0.25) | 59.32 | 0.68 | 1.25 | 1.27 | 1.07 | 1.44 | 2.86 | 2.69 | 2.33 |
| SADERF-FastFoodL | 36.02 (1.38) | 64.63 (0.18) | 76.99 (0.61) | 59.21 | 1.07 | 2.08 | 2.16 | 1.77 | 0.92 | 1.84 | 1.80 | 1.52 |
| OPRF-SGQ | 37.10 (0.23) | 61.25 (0.54) | 78.69 (0.54) | 59.01 | 0.67 | 1.26 | 1.25 | 1.06 | 1.36 | 2.71 | 2.56 | 2.21 |
| PosRF-ORF | 34.35 (5.96) | 60.30 (0.97) | 80.45 (0.22) | 58.37 | 0.56 | 1.05 | 1.06 | 0.89 | 1.17 | 2.31 | 2.10 | 1.86 |
| PosRF-FastFoodL | 33.46 (3.70) | 64.65 (0.36) | 76.95 (0.48) | 58.35 | 1.02 | 1.98 | 2.03 | 1.68 | 0.79 | 1.57 | 1.53 | 1.30 |
| SADERF-SORF | 33.30 (0.98) | 64.70 (0.36) | 74.71 (1.90) | 57.57 | 0.68 | 1.24 | 1.28 | 1.07 | 1.44 | 2.86 | 2.69 | 2.33 |
| PosRF-SGQ | 28.64 (7.54) | 62.38 (0.53) | 78.28 (0.20) | 56.43 | 0.56 | 1.05 | 1.05 | 0.89 | 1.17 | 2.31 | 2.10 | 1.86 |
| OPRF-SORF | 27.91 (3.26) | 64.76 (0.66) | 75.92 (1.74) | 56.20 | 0.67 | 1.26 | 1.24 | 1.06 | 1.36 | 2.71 | 2.56 | 2.21 |
| PosRF-SORF | 21.27 (6.65) | 62.99 (0.40) | 67.10 (1.11) | 50.45 | 0.56 | 1.05 | 1.05 | 0.89 | 1.17 | 2.31 | 2.10 | 1.86 |

The LRA covers tasks of different sequence lengths, difficulty, and objective and it is designed to evaluate efficient Transformers, of which Spectraformer belongs to. Specifically, we evaluate the models on three textual LRA tasks: ListOps (Nangia & Bowman, 2018), Text (Byte-Level Text Classification, using the IMDb review dataset) (Howard & Ruder, 2018), and Retrieval (Byte-Level Document Retrieval, using the ACL Anthology Network dataset) (Radev et al., 2013). We discuss the task characteristics in greater details in Section 4.2. All our models are experimented on NVIDIA V100 and 12 CPU with 20GB of memory. Our codebase is based on and includes the adapted or original implementation from Chen et al. (2021); Chowdhury et al. (2022); Liu et al. (2022); Choromanski et al. (2021); Likhosherstov et al. (2022; 2023); Wang et al. (2020); Xiong et al. (2021), Kitaev et al. (2020); Thomas et al. (2018); Zhou et al. (2021) (Apache License, Version 2.0).

We run our experiments on 5 different seeds due to computational limitation. The parameters are identical to Chen et al. (2021) and chosen to limit the parameter count to account for the marked training time of multiple models. All the parameters for Spectraformer variants are kept identical for fair comparison, with the only difference being their random feature component. Our code is implemented in Python 3.12 and Pytorch, we use the Transformer base from Chen et al. (2021). Our hyperparameters are available in Table 8. The model name convention is [component function]-[weight matrix] (e.g., OPRF-FastFood$_L$).

Our result in Table 3 shows that the top mean accuracy models, ranked in descending order, are: OPRF-FastFoodL, OPRF-MM, PosRF-MM, OPRF-ORF, SADERF-QMC. Our mean accuracy result indicates that novel Spectraformer combinations have outperformed the highlighted previous SOTA, i.e., OPRF-ORF and SADERF-ORF. However, we are aware that mean accuracy is insufficient in offering a comprehensive picture. Therefore, we first proceed with analyzing the statistics of the tasks before analyzing the significance of these results.

## 4.2 TASK CHARACTERISTICS

We first analyze the characteristics of the three textual classification LRA (Tay et al., 2021) tasks: ListOps (Nangia & Bowman, 2018), Text (Byte-Level Text Classification, using the IMDb review dataset) (Howard & Ruder, 2018), and Retrieval (Byte-Level Document Retrieval, using the ACL Anthology Network dataset) (Radev et al., 2013). We take into consideration three main characteristics:

- **Number of output classes**. ListOps is a task which inputs a sequence containing brackets and simple mathematical operators, with the output being a integer between 1 and 10, hence it has 10 classes. Text is a sentiment classification task on an IMDb review dataset with the class being either negative or positive, hence it has 2 classes. Retrieval inputs two sequences and the output is either False (meaning the sequences are not of the same document) or True (meaning the sequences are of the same document).

- **Maximum sequence size**. ListOps is 2K; Text is 4K; whilst Retrieval requires the processing of two sequences each of 4K length making the total maximum sequence size being 8K.

- **Required attention span**. This is the 'mean distance between the query token and the attended tokens, scaled by attention weights'. The average values for required attention span calculated by Tay et al. (2021) is: ListOps being 0.8, Text being 0.3, and Retrieval being 1.3.

Table 4: Mean Pearson correlation between model performance statistics and each task characteristic across weight matrices and component functions. Model performance statistics include 'acc' (accuracy), 'time' (training time), 'mem' (peak memory consumption). Task characteristics are 'size' (maximum sequence size), 'span' (required attention span), and 'classes' (number of output classes). Correlation is calculated for each weight matrix or component function and the average value over weight matrix or component function respectively displayed. Left is weight matrix $W$, right is component function $f$.

| $W$ | acc | time | mem |
|---|---|---|---|
| size | 0.93 | 0.74 | 0.64 |
| span | 0.37 | 0.02 | -0.10 |
| classes | -0.91 | -0.96 | -0.94 |

| $f$ | acc | time | mem |
|---|---|---|---|
| size | 0.93 | 0.56 | 0.60 |
| span | 0.36 | 0.02 | -0.10 |
| classes | -0.91 | -0.72 | -0.89 |

We then calculate Pearson correlation of accuracy, training time, and memory consumption against these characteristics. Our result is shown in Table 4. The table indicates that: (1) contrary to intuition, longer sequence size gives higher accuracy but this is not because of increasing required attention span and they somewhat increase training time and peak memory consumption; (2) increasing the number of classes decreases accuracy, training time, and peak memory consumption.

## 4.3 ANALYSIS OF COMBINATIONS

Table 5: Mean statistics across weight matrices for each task. Task is denoted on the top left cell of each table. Mean statistics include 'acc' (accuracy (%)), 'time' (training time (hour)), and 'mem' (peak memory consumption (GB)).

| ListOps | acc | time | mem |
|---|---|---|---|
| SORF | 27.49 | 0.64 | 1.32 |
| SGQ | 34.28 | 0.64 | 1.32 |
| FastFoodL | 35.68 | 1.05 | 0.86 |
| ORF | 36.59 | 0.64 | 1.32 |
| QMC | 37.39 | 0.64 | 1.32 |
| MM | 37.41 | 0.64 | 1.32 |

| Text | acc | time | mem |
|---|---|---|---|
| ORF | 60.28 | 1.19 | 2.63 |
| QMC | 60.92 | 1.19 | 2.63 |
| MM | 60.98 | 1.19 | 2.63 |
| SGQ | 62.03 | 1.19 | 2.63 |
| SORF | 64.15 | 1.18 | 2.63 |
| FastFoodL | 64.56 | 2.04 | 1.71 |

| Retrieval | acc | time | mem |
|---|---|---|---|
| SORF | 72.58 | 1.19 | 2.45 |
| FastFoodL | 77.22 | 2.10 | 1.67 |
| SGQ | 78.45 | 1.19 | 2.45 |
| QMC | 80.59 | 1.20 | 2.45 |
| ORF | 80.79 | 1.20 | 2.45 |
| MM | 80.93 | 1.20 | 2.45 |

We now explore the suitability of weight matrices and component functions depending on task characteristics. Table 5 shows the mean statistics across weight matrices for each task. We notice that weight matrices form two different groups which align with the required attention span differences. ListOps and Retrieval are tasks with high required attention span, ListOps being 0.8 and Retrieval being 1.3. In both these tasks, we observe that ORF, QMC, and MM perform comparatively against each other and a lot better against SORF, SGQ, and FastFoodL. However, the Text task, which has

a much lower required attention span of 0.3, has much higher performance from the SORF-SGQ-FastFoodL group, whilst ORF, QMC, and MM perform much poorly on this task. Based on this observation, we can conclude that ORF, QMC, and MM are suited for high required attention span; whilst SORF, SGQ, and FastFoodL are suited for low required attention span. Analyzing mean statistics per task across component functions do not show any task-specific trend (refer to Table 9 for more information).

Table 6: Summary statistics of component functions $f$ (top) and weight matrices $W$ (bottom) in mean accuracy (on the test set), mean time (training time (hour)), and mean memory (peak memory consumption (GB)). $f$ and $W$ entries are sorted by mean accuracy respectively.

|   |   | Accuracy (%) ↑ | Time (hour) ↓ | Memory (GB) ↓ |
|---|---|---|---|---|
| $f$ | SADERF | 59.17 | 1.19 | 2.19 |
| $f$ | OPRF | 59.02 | 1.18 | 2.08 |
| $f$ | PosRF | 57.20 | 1.02 | 1.77 |
| $W$ | MM | 59.78 | 1.01 | 2.13 |
| $W$ | QMC | 59.63 | 1.01 | 2.13 |
| $W$ | ORF | 59.22 | 1.01 | 2.13 |
| $W$ | FastFoodL | 59.15 | 1.73 | 1.41 |
| $W$ | SGQ | 58.25 | 1.01 | 2.13 |
| $W$ | SORF | 54.74 | 1.00 | 2.13 |

In analyzing efficiency, we follow the definition of efficient Transformers in Tay et al. (2022) in terms of memory and computation, the latter interpreted as computational cost. The overall trend of accuracy, time, and memory of component functions is that they increase linearly from PosRF, OPRF, to SADERF (see Table 6). Hence, when limited computational resource is a concern, PosRF is a better option. Training time and peak memory consumption relies more heavily on component functions than weight matrices, with the exception of FastFoodL. FastFoodL takes significantly less memory consumption at the expense of training time. The accuracy trend for weight matrices is as described above. Table 6 also shows that whilst FastFoodL generally does not work as well due to its low required attention span, its high performance in the task of Text which is low in required attention span and the anomalously performant OPRF-FastFoodL leads to a data skew in the final ranking in Table 3.

Comparing between combinations, we find that for ListOps, OPRF-ORF is the most accurate (38.34%), followed by OPRF-MM (38.08%) and OPRF-QMC (37.69%). OPRF-ORF, being the previous SOTA, is undoubtedly the best choice here since other models maintain the same training time and peak memory consumption with lower accuracy. For Text, the most accurate model is OPRF-SORF (64.76%), followed by SADERF-SORF (64.70%) and PosRF-FastFoodL (64.65%). These all outperform the previous SOTAs (OPRF-ORF and SADERF-ORF) by $4\%+$ whilst maintaining a diversity of training time and peak memory consumption options. Both OPRF-SORF and SADERF-SORF maintain the same level of training time and peak memory consumption as the SOTAs, PosRF-FastFoodL offers much less peak memory consumption at the expense of longer training time. For Retrieval, the most accurate model is SADERF-MM (81.13%), followed by OPRF-MM (81.09%) and SADERF-ORF (81.05%). Whilst these models only perform slightly better than the SOTA SADERF-ORF, OPRF-MM offers less peak memory consumption compared to SADERF-ORF.

## 5 CONCLUSION

We present Spectraformer, a framework for approximating and learning the attention kernel in the Transformer. Our paper generalizes past works and presents empirical findings on different component function and weight matrix combinations. We experiment with 18 combinations, of which 14 are novel. Novel combinations perform competitively against, if not more performant than the previous SOTA in the family for LRA textual tasks. We show empirically that the choice of performant weight matrix depends on the required attention span of the task. Our results and the extensibility of our work shows the viability of Spectraformer as a unifying random feature framework of Transformer.

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

# A APPENDIX

## A.1 FUTURE WORK, LIMITATIONS, AND BROADER IMPACT

Spectraformer provides an experimental framework for random features in the Transformer. However, we have not provided formal mathematical proofs for the bounds the these combinations due to the number of combinations discussed. We hope Spectraformer inspires future work to generalizes the proof in such a framework.

Spectraformer has covered a significant number of combinations, however, we have not covered all weight matrices and component functions in the literature. Spectraformer can serve as a foundation for further random feature Transformer research using the techiques not yet experimented with but covered in Appendices A.3, A.4, A.5.

Due to the significant number of experimented models, we are unable to perform hyper-parameter fine-tuning. Future work can benefit from extending the scope of the current research to other tasks, i.e., other non-textual LRA tasks (image classification, path-finder, and path-x) and short-context datasets/ tasks such as GLUE (Linzen et al., 2018). Transformer-based architecture can also benefit from Spectraformer. Past random feature Transformer works (Likhosherstov et al., 2022; 2023) have experimented with conformer and vision Transformer with ViT. Jeevan & Sethi (2022) provides a good foundation to extend our work. Spectraformer, although currently experimented in the Transformer setting, can also extend to non-parametric kernel classification. It could also benefit from ablation studies like the comparison variance in Chowdhury et al. (2022) or in Yao et al. (2023)

Given the complex nature of language models in the ethical, energy, and societal domains, our work should be used responsibly (Weidinger et al., 2021).s

## A.2 RELATED WORK

Spectraformer is based on several theoretical assumptions and choice of mathematical framework and approach. This section discusses alternatives to the choices of Spectraformer. Specifically, Appendix A.2.1 discusses alternative kernelized formulations to Section 2.1; Appendix A.2.2 discusses alternative kernel approximation techniques to random features in Section A.2.2. There are also some component functions and weight matrices which are not covered in the experiments. We introduce them in the related work section in Appendices A.3, A.4, A.5.

### A.2.1 ALTERNATIVE KERNELIZED ATTENTIONS

In addition to the kernelized formulation of attention as shown in Section 2.1, based on our literature survey, we have identified the following three alternative kernel-based formulations of attention.

**Nadaraya-Watson kernel estimator in integration form (Nguyen et al., 2022):** Given the Gaussian kernel, where the probability of a single variable and the joint probability of two variables are estimated using kernel density estimation with the Gaussian kernel, then we have Equation 12, leading to the formulation of FourierFormer.

$$
\begin{aligned}
\phi(\boldsymbol{\delta}) &= exp(-||\boldsymbol{\delta}||^2/2\sigma^2) \\
\hat{r}_n(\boldsymbol{k}) &= \int \frac{\boldsymbol{v}p(\boldsymbol{v},\boldsymbol{k})}{p(\boldsymbol{k})}dv = \frac{\sum_{j=1}^{N}\hat{\boldsymbol{v}}_j\phi(\boldsymbol{k}-\boldsymbol{k}_j)}{\sum_{j=1}^{N}\phi(\boldsymbol{k}-\boldsymbol{k}_j)} \\
\hat{r}_n(\boldsymbol{q}_i) &= \sum_{j=1}^{N}softmax(\boldsymbol{q}_i^T\boldsymbol{k}_j/\sigma^2)\hat{\boldsymbol{v}}_j
\end{aligned}
\tag{12}
$$

**Gaussian decomposition (Song et al., 2021):** the attention mechanism can also be decomposed into the Gaussian kernel directly from Equation 4 without appealing to the use of kernel estimator (Song et al., 2021). Equation 13 leads to Implicit Kernel Attention (IKA).

$$
A_i = \sum_{j=1}^{N} \frac{1}{Z_l(\boldsymbol{q}_i, \boldsymbol{K})} exp(\frac{-||\boldsymbol{q}_i - \boldsymbol{k}_j||_2^2}{2\sqrt{d_k}}) exp(\frac{||\boldsymbol{q}_i||_{p=2}^2 + ||\boldsymbol{k}_j||_{p=2}^2}{2\sqrt{d_k}})\boldsymbol{v}_j
\tag{13}
$$

**Non-local operation (Wang et al., 2018)** is popular decomposition of attention in computer vision. This is the basis for Vision Transformer and related work. Non-local operation itself a generalisation

of the non-local means used in image denoising (Buades et al., 2005). Non-local operation is essentially an operation that operates on the entire input range, instead of a specific window which is characteristic of convolutional operations (see Equation 14). Indeed, the attention vector output $\boldsymbol{y}$ can be defined using $f$ which is the softmax (see Equation 15) in the non-local operation.

$$\mathbf{y}_i = \frac{1}{C(\boldsymbol{x})} \sum_{\forall j} f(\boldsymbol{x}_i, \boldsymbol{x}_j) g(\boldsymbol{x}_j) \tag{14}$$

$$\mathbf{y} = softmax(\boldsymbol{x}^T \boldsymbol{W}_\theta^T \boldsymbol{W}_\theta \boldsymbol{x}) g(\boldsymbol{x}) \tag{15}$$

We will show that the non-local operation in Equation 14 is a generalisation of the Nadaraya-Watson kernel estimator. Specifically, when defining $C(\boldsymbol{x}) = \sum_l \mathcal{K}(\boldsymbol{q}_i, \boldsymbol{k}_l), f = \mathcal{K}(\boldsymbol{q}_i, \boldsymbol{k}_l), g = \boldsymbol{v}_j$, then we obtain the original Nadaraya-Watson kernel estimator in Equation 4. The variety of kernel decompositions of attention demonstrate the robustness of the kernel interpretation of attention and hence opening up an alley in attention improvement through the lens of kernel.

### A.2.2 KERNEL APPROXIMATION BEYOND RANDOM FEATURES

Kernel approximation is presented in Section 2.3 through the use of random features. However, there has been other kernel approximation techniques outside of random features. These include greedy basis selection (Smola & Schökopf, 2000), divide and conquer (Hsieh et al., 2014; Zhang et al., 2013), and nyström methods (Williams & Seeger, 2000) (which led to the development of Skyformer (Chen et al., 2021) and Nyströmformer (Xiong et al., 2021)).

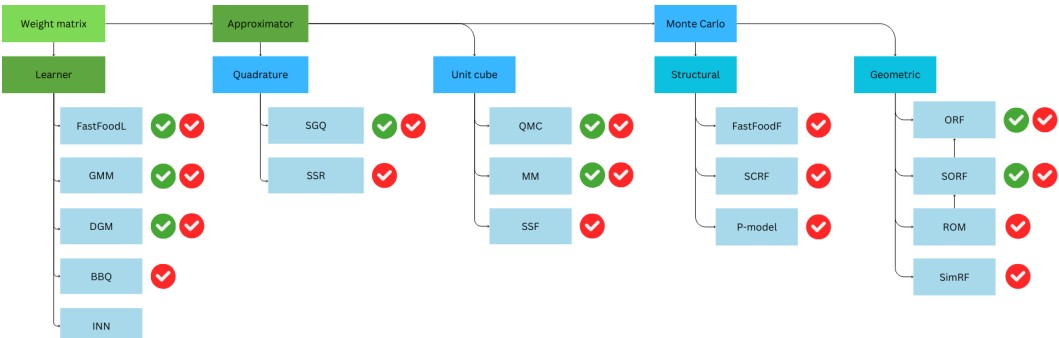

Figure 2: Spectraformer weight matrices. Red tick denotes techniques which have been explored in the literature. Green tick denotes techniques which are explored in this work. Please refer to Appendix A.3 for references.

### A.3 TECHNICAL DETAILS OF WEIGHT APPROXIMATORS

Spectraformer weight matrices are shown in Figure 2. Weight matrices are either approximators or learners. There are three approaches to approximators: Monte Carlo sampling, Unit-cube sampling, and quadrature. We have discussed these approaches in detail in Section 3.1. Here we provide a more specific classification schemes for each method.

- **Monte Carlo sampling**-based weight matrices are improvements over Base and subdivided into two types: structural and geometric methods. Structural methods decompose the Base random matrices $P$ into smaller matrices which reduce time and/ or space complexity. Structural matrices includes FastFood$_F$, SCRF and P-Model. Among these, we have explored the learner variant FastFood$_L$ of FastFood. Geometric methods enforce certain geometrical couplings along some dimensions in the construction of random matrices to reduce the approximation error. Geometric methods include: ROM (ORF, SORF) and SimRF. Among these, we have explored the main ROM variants - ORF and SORF.

- **Unit-cube sampling**-based weight matrices reformulate the Bochner integral in Equation 9 as Equation A.3 with uniform and independent points. This approach includes QMC, MM, and SSF. We have not covered SSF in our experiments.

- **Quadrature**-based weight matrices estimates the intractable integral using non-uniform weights with deterministic rules. There are two approaches: SGQ and SSR. We have not covered SSR in our experiments.

We now explore these weight matrices in the order specified above.

**Base** (Rahimi & Recht, 2007) is the direct sampling from a distribution $P_{ij} \sim p$, corresponding to the kernel $\mathcal{K}$ without any additional adjustment, and in approximating the RBF, $P = G$ where $G$ is the normal distribution.

$$W_{Base} = \frac{1}{\sigma} \boldsymbol{P} \tag{16}$$

**FastFood$_F$** (Le et al., 2013) makes use of Hadamard and diagonal matrices to speed up Gaussian matrices $G$ constructions for RFF in $O(n \log d)$ (100 times faster) with $O(n)$ space (1000 times less space) given $n \geq d$ against Base. However, FastFood$_F$ increases variance and approximation error and decreases the concentration bound.

$$W_{FastFood_L} = \frac{1}{\sigma} \boldsymbol{SHG\Pi HB} \tag{17}$$

where $\boldsymbol{H}$ is the Walsh-Hadamard matrix, $\boldsymbol{\Pi} \in \{0,1\}^{d*d}$ is a permutation matrix, and $\boldsymbol{S}, \boldsymbol{G}, \boldsymbol{B}$ are diagonal random matrices, with the diagonal entries being $\{+-1\}$ entries on $\boldsymbol{B}$, random Gaussian entries on $\boldsymbol{G}$, and a random scaling matrix on $\boldsymbol{S}$, $\boldsymbol{S}_{ii} = s_i ||\boldsymbol{G}||_{Frob}^{-1/2}, s_i \sim (2\pi)^{\frac{-d}{2}} \boldsymbol{A}_{d-1}^{-1} r^{d-1} e^{-\frac{r^2}{2}}$.

**SCRF** (Signed Circulant Matrix Projection) (Feng et al., 2015) maintains a high matrix construction speed and $O(s)$ space ($s = t * d$) whilst maintaining the variance of Base, an improvement over FastFood$_F$. It does this by making use of circulant matrices.

$$\boldsymbol{W}_{SCRF} = [\boldsymbol{P}^{(1)}, \boldsymbol{P}^{(2)}, ..., \boldsymbol{P}^{(t)}], t = d|D \tag{18}$$

where $\boldsymbol{P}^{(i)}$ is a signed circulant Gaussian matrix, with the first column of circulant random matrix $\boldsymbol{C}^{(i)[d]}$ of $\boldsymbol{P}^{(i)}$ drawn randomly from a Gaussian distribution $\mathcal{N}(\boldsymbol{0}, \boldsymbol{I}/\sigma^2)$. The circulant matrix via Discrete Fourier Transform can be defined as:

$$\begin{aligned} \boldsymbol{C} &= \frac{1}{m} \boldsymbol{F}^* diag(\boldsymbol{Fc}) \boldsymbol{F} \\ \boldsymbol{F} &= [e^{i\frac{2\pi}{m}kn}]_{k,n=0}^{m-1} \end{aligned} \tag{19}$$

**P-Model** (Choromanski & Sindhwani, 2016) generalizes over both circulant matrices (including SCRF), FastFood$_F$, along with other Gaussian and semi-Gaussian structured matrix methods.

$$\boldsymbol{W}_{P_{model}} = [\boldsymbol{g}^T \boldsymbol{P_1}, \boldsymbol{g}^T \boldsymbol{P_2}, ..., \boldsymbol{g}^T \boldsymbol{P_s}]^T \tag{20}$$

where $\boldsymbol{g} \in \mathbb{R}^a$ is a Gaussian vector and a sequence of $m$ $\boldsymbol{P_i} \in \mathbb{R}^{a*d}$ matrices with L2 norm columns, where $P_i$ represents a P-model. Semi-Gaussian structured matrices including Toeplitz, Hankel, and other Toeplitz-like matrices are an open area of research for random matrices.

**ORF** (Orthogonal RF) (Yu et al., 2016) decreases the approximation error significantly compared to Base. This is done via replacing $G$ with a properly scaled random orthogonal matrix. However, generating orthogonal matrices become costly quickly as the number of dimensions increases.

$$\boldsymbol{W}_{ORF} = \frac{1}{\sigma} \boldsymbol{SQ} \tag{21}$$

where $\boldsymbol{Q}$ is a uniformly distributed random orthogonal matrix (on the Stiefel manifold) obtained from the QR decomposition of $\boldsymbol{G}$, the set of rows of $\boldsymbol{Q}$ forming a basis in $\mathbb{R}^d$, and $\boldsymbol{S}$ is a diagonal matrix with entries sampled i.i.d from the $\chi$-distribution with $d$ degrees of freedom, thus making the rows of $\boldsymbol{SQ}$ and $G$ identically distributed.

**SORF** (Structured ORF) (Yu et al., 2016) decreases time and space complexity of ORF (from $O(d^2)$

to $O(d \log d)$ with almost no extra memory cost) by imposing structure on the orthogonal matrices, inspired by structural methods. SORF is unbiased with large $d$. We replace $\boldsymbol{S}$ with $\sqrt{d}$ and $\boldsymbol{Q}$ with structured matrix $\boldsymbol{HD_1HD_2HD_3}$

$$\boldsymbol{W}_{SORF} = \frac{\sqrt{d}}{\sigma} \boldsymbol{HD_1HD_2HD_3} \tag{22}$$

where $\boldsymbol{D}_i \in \mathbb{R}^{d*d}, i = 1, 2, 3$ is diagonal 'sign-flipping' matrices with diagonal entries sampled from the Rademacher distribution, $\boldsymbol{H}$ is the normalized Walsh-Hadamard matrix.

**ROM** (Random Orthogonal Embeddings) (Choromanski et al., 2017) generalizes SORF ($t = 3$) to:

$$\boldsymbol{W}_{ROM} = \frac{\sqrt{d}}{\sigma} \prod_{i=1}^{t} \boldsymbol{SD}_i \tag{23}$$

where $\boldsymbol{S}$ is a class of $L_2$-normalized versions of Kronecker product matrices, of which the Hadamard $\boldsymbol{H}$ is a representative of, and $\boldsymbol{D}$ being independent diagonal matrices. When $\boldsymbol{S} = \boldsymbol{H}$ and $t = 3$, we obtain SORF used in our experiments. Another ROM variant is S-Hybrid, which we do not experiment with, given as

$$\boldsymbol{W}_{S_{Hybrid}} = \boldsymbol{SD}_k^{(\mathcal{U})} \prod_{i=1}^{k-1} \boldsymbol{SD}_i^{(\mathcal{R})} \tag{24}$$

where $\boldsymbol{D}_k^{(\mathcal{U})}$ is a diagonal matrix with i.i.d. $Unif(\mathcal{S}^1)$ on the unit circle of $\mathbb{C}$. When being used in the component function, instead of calculating $\phi(\boldsymbol{x})^T \phi(\boldsymbol{y})$ in Equation 10. We model the equation as $Real(\overline{\phi(\boldsymbol{x})}^T \phi(\boldsymbol{y}))$.

**SimRF** (Simplex RF) (Reid et al., 2023) is developed to be a positive RF with the most optimal solution, the MSE of which is the lowest in the geometric methods. This is an alternative to the ROM-based geometric methods. Two variants of SimRF are: weight-independent and weight-dependent. SimRF is defined as:

$$\boldsymbol{W}_{simp} = \boldsymbol{DSR} \tag{25}$$

where $\boldsymbol{D} \in \mathbb{R}^{d*d} = diag(\boldsymbol{\omega}_i), \boldsymbol{R} \in \mathbb{R}^{d*d}$ being a random orthogonal matrix drawn from Haar measure on $O(d)$ and $\boldsymbol{S} \in \mathbb{R}^{d*d}$ with row $\boldsymbol{s}_i$:

$$\boldsymbol{s}_i = \begin{cases} \sqrt{\frac{d}{d-1}}\boldsymbol{e}_i - \frac{\sqrt{d}+1}{(d-1)^{3/2}}(1,...,1,0)^T & 1 \le i < d \\ \frac{1}{\sqrt{d-1}}(1,1,...,1,0)^T & i = d \end{cases} \tag{26}$$

being unit vectors which are manifestly normalized and subtend obtuse angles. SimRF variants are defined based on how $\boldsymbol{\omega}_i$ is constructed. For SimRF$_{\text{indep}}$, $\boldsymbol{\omega}_i \sim \chi_d$. For Sim$_{\text{dep}}$, we permit the random vector direction $\{\hat{\boldsymbol{\omega}}_i\}$ to be correlated with norms $\{\boldsymbol{\omega}_i\}$:

$$\boldsymbol{\omega}_i = -\frac{\sum_{j \neq i} \omega_j}{||\sum_{j \neq i} \omega_j||_2} \boldsymbol{\omega}_i, i = 1,...,d \tag{27}$$

**QMC** (Quasi-Monte Carlo) (Avron et al., 2016) evaluates on a low discrepancy sequence (e.g., Halton, Sobol' Faure, and Niederreiter) of points instead of random points in Monte Carlo. Although the approximation error is only reduced minimally, QMC has been shown to perform better than MC in high dimensions and does not have undesirable clustering effect. To calculate QMC, we first assume that $p(\boldsymbol{x}) = \prod_{j=1}^{d} p_j(\boldsymbol{x}_j)$ factorizes with respect to the dimensions with $p_j(.)$ being a univariate density function. Then we define:

$$\Phi^{-1}(\boldsymbol{t}) = \left(\Phi_1^{-1}(\boldsymbol{t}_1), \ldots, \Phi_d^{-1}(\boldsymbol{t}_d)\right) \in \mathbb{R}^d \tag{28}$$

where $\Phi_j$ being the cumulative distribution function of $p_j$, $\boldsymbol{t}_1, \boldsymbol{t}_2, ..., \boldsymbol{t}_s \in [0,1]^d$ being a low discrepancy sequence, and $\boldsymbol{\omega}_i = \Phi^{-1}(\boldsymbol{t}_i)$. We can thus transform the integral on $\mathbb{R}^d$ in Equation 9 to an integral on the unit cube $[0,1]^d$ as

$$\mathcal{K}(\boldsymbol{x} - \boldsymbol{x}') = \int_{[0,1]^d} \exp\left(\mathrm{i}(\boldsymbol{x} - \boldsymbol{x}')^\top \Phi^{-1}(\boldsymbol{t})\right) \mathrm{d}\boldsymbol{t}$$

The weight matrix then can be defined as:

$$\boldsymbol{W}_{\text{QMC}} = [\Phi^{-1}(\boldsymbol{t}_1), \Phi^{-1}(\boldsymbol{t}_2), \ldots, \Phi^{-1}(\boldsymbol{t}_s)]^\top \in \mathbb{R}^{s \times d} \ .$$

QMC can be further improved by a sub-grouped based rank-one lattice construction which improved complexity (Lyu et al., 2020).

**MM** (Moment Matching) (Shen et al., 2017; Liu et al., 2022) improves over QMC by removing the undesirable clustering effect and having the same approximation error with less features. This is done by replacing $\Phi^{-1}$ with a moment matching scheme $\widetilde{\Phi}^{-1}$:

$$\boldsymbol{W}_{\text{MM}} = [\widetilde{\Phi}^{-1}(\boldsymbol{t}_1), \widetilde{\Phi}^{-1}(\boldsymbol{t}_2), \ldots, \widetilde{\Phi}^{-1}(\boldsymbol{t}_s)]^\top \in \mathbb{R}^{s \times d}$$

where $\widetilde{\Phi}^{-1}(\boldsymbol{t}_i) = \tilde{\boldsymbol{A}}^{-1}(\Phi^{-1}(\boldsymbol{t}_i) - \tilde{\boldsymbol{\mu}})$ can be constructed using moment matching with sample mean $\tilde{\boldsymbol{\mu}} = \frac{1}{s}\sum_{i=1}^{s} \Phi^{-1}(\boldsymbol{t}_i)$ and the square root of the sample covariance matrix $\tilde{\boldsymbol{A}}$ satisfying $\tilde{\boldsymbol{A}}\tilde{\boldsymbol{A}}^\top = \text{Cov}(\Phi^{-1}(\boldsymbol{t}_i) - \tilde{\boldsymbol{\mu}})$.

**SSF** (Spherical Structured Features) (Lyu, 2017) improves over QMC by including rotation-invariant kernel as well as in terms of complexity. Rotation-invariant property suggests the construction of feature maps using spherical equal weight approximation using Riesz s-energy on a d-dimensional sphere $\mathbb{S}^d := \{\boldsymbol{x} \in \mathbb{R}^{d+1} | \|\boldsymbol{x}\|_2 = 1\}$. Specifically, we construct $\{\boldsymbol{v}_i\}_{i=1}^s$ asymptotically uniformly distributed on the sphere and obtain $\boldsymbol{V} := [\boldsymbol{v}_1, \boldsymbol{v}_2, \ldots, \boldsymbol{v}_s] \in \mathbb{S}^{(d \times s)}$, that $\Phi^{-1}(t)$ uses the one-dimensional QMC point, we have the weight matrix:

$$\boldsymbol{W}_{\text{SSF}} = [\Phi^{-1}(t)\boldsymbol{v}_1, \Phi^{-1}(t)\boldsymbol{v}_2, \ldots, \Phi^{-1}(t)\boldsymbol{v}_s]^\top \in \mathbb{R}^{s \times (d+1)},$$

**SGQ** (Sparse Grid Quadrature) (Dao et al., 2017) evolves from GQ. GQ (Gaussian Quadrature) (Dao et al., 2017) assumes that the kernel $k$ factorizes with respect to the dimensions and thus can be approximated by a one-dimensional Gaussian quadrature rule. However the total number of points $s$ scale exponentially with the dimensions. However GQ suffers form the curse of dimensionality. This is alleviated using Smolyak rule resulting in SGQ. Assuming the third-degree SGQ using symmetric univariate quadrature points $\{-\hat{p}_1, 0, \hat{p}_1\}$ with weights $(\hat{a}_1, \hat{a}_0, \hat{a}_1)$, then we have:

$$\boldsymbol{W}_{\text{SGQ}} = [\boldsymbol{0}_d, \hat{p}_1\boldsymbol{e}_1, \ldots, \hat{p}_1\boldsymbol{e}_d, -\hat{p}_1\boldsymbol{e}_1, \ldots, -\hat{p}_1\boldsymbol{e}_d]^\top \in \mathbb{R}^{(2d+1) \times d}$$

given that $_i$ is the d-dimensional standard basis vector with the i[th] element being 1.

**SSR** (Spherical-Radial Rules) (Munkhoeva et al., 2018) transforms Equation 10 into a double integral over a hyper-sphere and the real line. This leads to the following approximation:

$$\boldsymbol{W}_{\text{SSR}} = \boldsymbol{\vartheta} \otimes \begin{bmatrix} (\boldsymbol{Q}\boldsymbol{V})^\top \\ -(\boldsymbol{Q}\boldsymbol{V})^\top \end{bmatrix} \in \mathbb{R}^{(2d+1) \times d},$$

where $\boldsymbol{\vartheta} = [\vartheta_1, \vartheta_2, \ldots, \vartheta_s]$, $\boldsymbol{V} = [\boldsymbol{v}_1, \boldsymbol{v}_2, \ldots, \boldsymbol{v}_{d+1}]$, $\vartheta \sim \chi(d+2)$ and $\{\boldsymbol{v}_i\}_{i=1}^{d+1}$ being vertices of a unit regular d-simplex randomly rotated by $\boldsymbol{Q}$ (a random orthogonal matrix).

### A.4 TECHNICAL DETAILS OF WEIGHT LEARNERS

We give a non-mathematical definition of weight matrix learner in Section 3.1: weight matrix learner is any function which parameterize distributions. This allows for the discovery and future experimentation with other potential weight matrix learners in Spectraformer and random feature method in general. Weight matrix learner comes from a long line of work in kernel methods, namely, kernel learning via random features. Typically, a component function $f$ is chosen and a weight matrix $\boldsymbol{W}$ is produced via kernel approximation, providing us with the feature map $\phi$. $\phi$ can then be combined with input $\boldsymbol{x}$ and weight $\boldsymbol{v}$ to fit the learning objective. Kernel learning, however, imposes an additional objective: learning the weight matrix $\boldsymbol{W}$.

The approach to the dual objectives separates the kernel learning methods into two-stage and one-stage methods (Liu et al., 2022). Two-stage methods (e.g., Yu et al. (2015); Wilson & Adams (2013);

Bullins et al. (2018)) solves the dual objective separately: $W$ is typically first learned via a kernel alignment scheme (Wang et al., 2015), then we solve for $v$. One-stage methods (e.g., Chowdhury et al. (2022); Tompkins et al. (2019)), on the other hand, solves the dual objective in parallel: $p(.)$ corresponding to $W$ is parameterized then $v$ is solved typically. It is not plausible to apply kernel alignment (Wang et al., 2015) on Transformer architectures. Therefore, only one-stage methods can be considered in the context of weight matrix learner. Two kernel learning methods, GMM and FastFood$_L$, have been experimented in the non-Transformer setting (Yang et al., 2015), then adapted into the Transformer setting (Chowdhury et al., 2022). DGM (Deep Generative Model) has also been effective in modeling distributions (Kingma & Welling, 2013). It too has been adapted by Chowdhury et al. (2022) for the Transformer setting. Due to the superior performance of FastFood$_L$ with this technique, we have only experimented with this technique. We now explore them.

**FastFood**$_L$ (see Equation 17) is proposed to be learnable by Yang et al. (2015) with two variations: FSARD (scaling matrix $S$ previously sampled from chi-squared is made learnable) and FSGBARD (optimizes the marginal likelihood with respect to the diagonal matrices $G$ and $B$). It is further adapted by Chowdhury et al. (2022) to make $S$ or $S, G, B$ learnable parameters. In our experiments, we maintain this set up.

**GMM** (Gaussian Mixture Model) (Wilson & Adams, 2013; Oliva et al., 2016) is a known universal approximator of probability distribution. Specifically, we can consider $W_{GMM}$ being composed of $s$ components, with each component $c$ (with $C$ total components) being a Gaussian with mean vector $\mu_c$ and covariance matrix $\Sigma_c$ and component weight $\pi_c$. $p(\omega)$ can then be approximated as $p(\omega) = \sum_{c=1}^{C} \pi_c \mathcal{N}(\mu_c, \Sigma_c)$ (Chowdhury et al., 2022). Assuming $\pi_c = \frac{1}{C}$, the weight matrix with learnable weights $\Sigma$ and $\mu$ can be parameterized as:

$$W_{GMM} = [\Sigma_1 n + \mu_1, ..., \Sigma_C n + \mu_C]^T, n \sim \mathcal{N}(\mathbf{0}, \mathbf{I}) \tag{29}$$

Another formulation of GMM is proposed in Yang et al. (2015) in combination with FastFood, albeit it extends beyond constructing the weight matrix.

**DGM** (Deep Generative Model) (Kingma & Welling, 2013) allows for a generic distribution modeling using neural network. DGM is initially proposed to make differentiating sampling processes possible via the reparametrization trick, where the generator $g$ creates samples from the target distribution $\omega_m = g(n_m)$ via transforming from a noise distribution $n_m \sim p_0(.)$.

In addition to the techniques discussed above, there are also some other prominent one-stage methods (e.g., INN, BBQ). INN (Invertible Neural Network) (Xu et al., 2021) is another universal approximator of probability distribution, consisting of invertible operations transforming samples from a known distribution to a different more complex distribution. BBQ (Black Box Quantile) (Tompkins et al., 2019) follows the QMC scheme and models $\Phi^{-1}$ using parameterized quantile function.

### A.5   TECHNICAL DETAILS OF COMPONENT FUNCTIONS

Component functions discussed in Section 3.2 all belong to CDERF. However, there are other component functions including DIRF and HRF (see Figure 3) which we do not explore in our experiments. We now investigate the theoretical details of these component functions, the CDERF family is introduced in their order of discovery.

**CDERF** (Complex dense exponential random feature) is given in Equation 30, where $\omega, x \in \mathbb{R}^d$, $p = \mathcal{N}(0,1)^d$, $k \in \{1,2\}$ and $\mathbb{S}_{d_c}$ being a set of $d * d$ complex symmetric matrices. The parameters and constraints of different CDERF functions are specified in Table 7:

$$f_{DE}^{(k)}(\omega, x) = Real(D \exp(\omega^T A \omega + \omega^T B^{(k)} x + x^T C^{(k)} x)) \tag{30}$$

**TrigRF** (Trigonometric RF) (Rahimi & Recht, 2007) is the base RFF implementation.

$$\begin{aligned} f_{TrigRF}^{(1)}(\omega, x) &= \sqrt{2} \cos(\omega^T x + b) \\ f_{TrigRF}^{(2)}(\omega, y) &= \sqrt{2} \sin(\omega^T y + b) \\ b &\sim Uniform(0, 2\pi) \end{aligned} \tag{31}$$

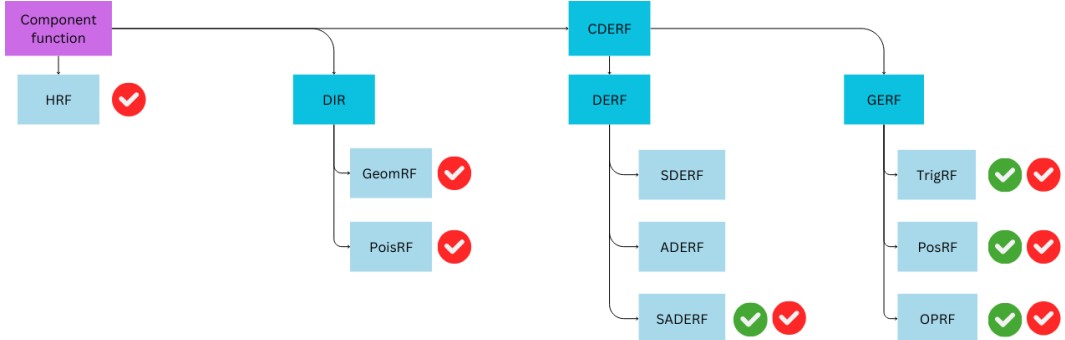

Figure 3: Spectraformer component functions. Red tick denotes techniques which have been explored in the literature. Green tick denotes techniques which are explored in this work. Please refer to Appendix A.5 for references.

Table 7: CDERF component functions

| | $A$ | $B^{(1)}$ | $B^{(2)}$ | $C^{(1)}$ | $C^{(2)}$ | $D$ | $x$ | $y$ |
|---|---|---|---|---|---|---|---|---|
| **CDERF** | $\mathbb{S}_{d_C}$ | $\mathbb{C}^{d*d}$ | | $\mathbb{C}^{d*d}$ | | $\mathbb{R}$ | $x$ | $y$ |
| **DERF** (Likhosherstov et al., 2023) | $\mathbb{S}_d$ | $\mathbb{R}^{d*d}$ | | $\mathbb{R}^{d*d}$ | | $\mathbb{R}$ | $x$ | $y$ |
| **SADERF** (Likhosherstov et al., 2023) | $A_{GE}\boldsymbol{I}_d$ | $B_{GE}\boldsymbol{I}_d$ | | $C_{GE}\boldsymbol{I}_d$ | | $D_{GE}$ | $\Psi x$ | $\Psi^{-1}y$ |
| **GERF** (Likhosherstov et al., 2022) | $A_{GE}\boldsymbol{I}_d$ | $B_{GE}^{(1)}\boldsymbol{I}_d$ | $B_{GE}^{(2)}\boldsymbol{I}_d$ | $C_{GE}\boldsymbol{I}_d$ | | $D_{GE}$ | $x$ | $y$ |
| **TrigRF** (Rahimi & Recht, 2007) | $0\boldsymbol{I}_d$ | $i\boldsymbol{I}_d$ | $-i\boldsymbol{I}_d$ | $0\boldsymbol{I}_d$ | | 1 | $x$ | $y$ |
| **PosRF** (Choromanski et al., 2021) | $0\boldsymbol{I}_d$ | $1\boldsymbol{I}_d$ | | $-1\boldsymbol{I}_d$ | | 1 | $x$ | $y$ |
| **OPRF** (Likhosherstov et al., 2022) | $0\boldsymbol{I}_d$ | $1\boldsymbol{I}_d$ | | $-1\boldsymbol{I}_d$ | | 1 | $x$ | $y$ |

The use of the trigonometric sine and cosine functions leads to unstable behavior when the inputs have negative dimension-values. This can be further exacerbated when the values TrigRF try to approximate are close to 0 (since most values are of low significance). This causes the variance to approach infinity (Choromanski et al., 2021; Likhosherstov et al., 2022). Therefore, we do not want to use TrigRF to perform kernel approximation in attention.

**PosRF** (Positive RF) (Choromanski et al., 2021) fixes the problem of TrigRF by enforcing positive component function output in the softmax. The variance of PosRF (in contrast to the variance of TrigRF approaching infinity) approaches 0 as the approximated value of the Softmax kernel approaches 0. PosRF has two forms: Positive RF Base (PosRF-B) (see Equation 32) and Positive RF Hyperbolic (PosRF-Hyp), which is multi-component, i.e., Equation 11: $l = 2$, (see Equation 33).

$$f_{PosRF_B}(\boldsymbol{\omega}, \boldsymbol{x}) = \exp(\boldsymbol{\omega}^T\boldsymbol{x} - \frac{||\boldsymbol{x}||^2}{2}) \tag{32}$$

$$f_{1,PosRF_{Hyp}}(\boldsymbol{\omega}, \boldsymbol{x}) = \exp(\boldsymbol{\omega}^T\boldsymbol{x} - ||\boldsymbol{x}||^2)$$
$$f_{2,PosRF_{Hyp}}(\boldsymbol{\omega}, \boldsymbol{x}) = \exp(-\boldsymbol{\omega}^T\boldsymbol{x} - ||\boldsymbol{x}||^2) \tag{33}$$

We only use PosRF-B in our experiments.

**GERF** (Generalized exponential RF) (Likhosherstov et al., 2022) generalizes both TrigRF and PosRF with Equation 34.

$$f_{GERF}^{(1)}(\boldsymbol{\omega}, \boldsymbol{x}) = D\exp(A||\boldsymbol{\omega}||^2 + B\boldsymbol{\omega}^T\boldsymbol{x} + \boldsymbol{C}||\boldsymbol{x}||^2)$$
$$f_{GERF}^{(2)}(\boldsymbol{\omega}, \boldsymbol{y}) = D\exp(\boldsymbol{A}||\boldsymbol{\omega}||^2 + sB\boldsymbol{\omega}^T\boldsymbol{y} + \boldsymbol{C}||\boldsymbol{y}||^2)$$

$$Re(1 - 4A) > 0, \boldsymbol{B} = \sqrt{s(1 - 4\boldsymbol{A})}$$
$$\boldsymbol{C} = -(s+1)/2, D = (\sqrt[4]{1 - 4\boldsymbol{A}})^d$$
$$\boldsymbol{A} \in \mathbb{C}, s \in \{-1, +1\} \tag{34}$$

where $\sqrt{\cdot}$ and $\sqrt[n]{\cdot}$ denoting a principal root with a complex argument

**OPRF** (Optimized positive RF) (Likhosherstov et al., 2022) is the solution to the minimization of the variance of GERF. Specifically it is defined as Equation 34 with $s = +1$, $||\boldsymbol{x} + \boldsymbol{y}||^2 > 0$, and $\boldsymbol{A} \in \mathbb{R}$ defined in terms of $p^*$ as:

$$\boldsymbol{A} = (1 - 1/p^*)/8$$
$$p^* = (\sqrt{(2||\boldsymbol{x} + \boldsymbol{y}||^2 + d)^2 + 8d||\boldsymbol{x} + \boldsymbol{y}||^2} - 2||\boldsymbol{x} + \boldsymbol{y}||^2 - d) \tag{35}$$
$$/(4||\boldsymbol{x} + \boldsymbol{y}||^2)$$

Whilst PosRF is not bounded, OPRF is. OPRF can provide $e^{60} \times$ variance reduction in estimating the Softmax compared to TrigRF.

**DERF** (Dense-exponential random features) (Likhosherstov et al., 2023) extends GERF and replace $\boldsymbol{A}, \boldsymbol{B}, \boldsymbol{C}$ with dense matrices. DERF is CDERF when $\boldsymbol{B}, \boldsymbol{C}$ are in the real instead of the complex plane. Minimizing the variance of DERF leads to two approaches: ADERF and SDERF. However both these approaches rely on SVD and eigen decompositions which are not extensively supported on GPU and deep learning libraries. Therefore SADERF is proposed.

**SADERF** (simplified ADERF) (Likhosherstov et al., 2023) is a special case of ADERF and extends GERF, requiring only basic unary operations in addition.

$$f_{SADE}^{(1)}(\boldsymbol{\omega}, \boldsymbol{x}) = f_{GE}^{(1)}(\boldsymbol{\omega}, \Psi \boldsymbol{x}), f_{SADE}^{(2)}(\boldsymbol{\omega}, \boldsymbol{y}) = f_{GE}^{(2)}(\boldsymbol{\omega}, \Psi^{-1} \boldsymbol{y})$$
$$\Psi_{l,l}^* = (\sum_j (\boldsymbol{y}_l^{(i)})^2 / \sum_i (\boldsymbol{x}_l^{(i)})^2)^{1/4} \tag{36}$$

**DIRF** (Discretely-induced random features) (Likhosherstov et al., 2022) are based on the assumption of $p(\boldsymbol{\omega})$ being a discrete distribution with $\boldsymbol{\omega}_1, ..., \boldsymbol{\omega}_d$ being i.i.d., $\mathbb{P}(\boldsymbol{\omega}_l = k) = p_k$, $\sum_{k=0}^{\infty} p_k = 1$, $p_k > 0, k \in \{0\} \cup \mathbb{N}$. By making use of the Taylor series, we can define DIRF as:

$$f_{DI}(\boldsymbol{\omega}, \boldsymbol{x}) = \exp(-\frac{||\boldsymbol{x}||^2}{2}) \prod_{l=1}^{d} \boldsymbol{x}_i^{\boldsymbol{\omega}_l} (\boldsymbol{\omega}_l!)^{\frac{-1}{2}} p_{\boldsymbol{\omega}_l}^{\frac{-1}{2}} \tag{37}$$

DIRF has two variations: PoisRF (Poisson RF) which requires $p(.)$ to be the Poisson distribution and GeomRF (Geometric RF) which requires $p(.)$ to be the geometric distribution. Since the many weight matrices introduced in Section 3.1 do not readily approximate these distributions, we decide to leave DIRF out of the paper.

**HRF** (Hybrid RF) (Choromanski et al., 2022) combines multiple base estimators, similar to multiple kernel learning methods (Bach et al., 2004), in order to provide the most accurate approximation in regions of interest. Choromanski et al. on solving softmax approximation using HRF. HRF is constructed using a weighted combination $\epsilon$ of $p + 1$ base estimators $\hat{SM}^k(\boldsymbol{x}, \boldsymbol{y})$ and $\lambda$-coefficients, $\Lambda p$ binary estimators $\lambda^k : \mathbb{R}^d \times \mathbb{R}^d \to [0, 1]$, both constructed independently:

$$\hat{SM}^{\epsilon, \Lambda} = \sum_{k=1}^{p} \hat{\lambda}^k(\boldsymbol{x}, \boldsymbol{y}) \hat{SM}^k(\boldsymbol{x}, \boldsymbol{y}) + (1 - \sum_{k=1}^{p} \hat{\lambda}^k(\boldsymbol{x}, \boldsymbol{y})) \hat{SM}^{p+1}(\boldsymbol{x}, \boldsymbol{y}) \tag{38}$$

Since HRF requires the base estimators $SM(\boldsymbol{x}, \boldsymbol{y})$ to be unbiased, the easiest solution is the complex exponential $\hat{SM}_m^{cexp}$ which can be derived directly from the expectation of the softmax as follows:

$$\hat{SM}_m^{cexp}(\boldsymbol{x}, \boldsymbol{y}) = \psi_A^m(\boldsymbol{x})^T \psi_{(\boldsymbol{A}^T)^{-1}}^m(\boldsymbol{y})$$
$$\psi_{\boldsymbol{M}}^m(u) = \frac{1}{\sqrt{m}} \exp(-\frac{(\boldsymbol{M}u)^2}{2})$$
$$(\exp(\boldsymbol{\omega}_i^T \boldsymbol{M} u), ..., \exp(\boldsymbol{\omega}_m^T \boldsymbol{M} u))^T, \boldsymbol{\omega}_i \sim \mathcal{N}(\boldsymbol{0}, \boldsymbol{I}_d) \tag{39}$$

To construct $\lambda$-coefficients, three methods are proposed: simultaneous accurate approximation of both small and large values, Gaussian Lambda Coefficients, and adaption to data admitting clustering structure.

## A.6 EXPERIMENTAL DETAILS IN SPECTRAFORMER

Table 8: Hyper-parameters for Spectraformer, following the setting from Chen et al. (2021) due to limited computational power. $L$ stands for ListOps, $T$ for Text, and $R$ for Retrieval. The seed that can be used to reproduce our experiments is 42.

|  | L | T | R |
|---|---|---|---|
| Embedding dim. | 64 | 64 | 64 |
| Transformer dim | 64 | 64 | 64 |
| Hidden dim | 128 | 128 | 128 |
| Head dim | 32 | 32 | 32 |
| Num. heads | 2 | 2 | 2 |
| Num. layers | 2 | 2 | 2 |
| Vocabulary size | 32 | 512 | 512 |
| Sequence length | 2000 | 4000 | 4000 |
| Dropout rate | 0.1 | 0.1 | 0.1 |
| Att. dropout rate | 0.1 | 0.1 | 0.1 |
| Pooling mode |  | mean |  |
| Batch size | 32 | 32 | 16 |
| Learning rate | 0.0001 | 0.0001 | 0.0002 |
| Warmup steps | 1000 | 80 | 800 |
| Learning rate decay |  | linear |  |
| Weight decay | 0 | 0 | 0 |
| Evaluation freq. | 500 | 500 | 1000 |
| Num. epochs | 50k | 50K | 50k |
| Num. init steps | 1k | 3K | 3k |
| Num. eval steps | 62 | 200 | 300 |
| Patience | 10 | 10 | 10 |
| Num. features | 128 | 128 | 128 |

Table 9: Mean statistics across component functions for each task. Task is denoted on the top left cell of each table. Mean statistics include 'acc' (accuracy), 'time' (training time), and 'mem' (peak memory consumption).

| ListOps | acc | time | mem |
|---|---|---|---|
| **PosRF** | 31.98 | 0.63 | 1.10 |
| **OPRF** | 36.11 | 0.74 | 1.28 |
| **SADERF** | 36.33 | 0.75 | 1.35 |

| Text | acc | time | mem |
|---|---|---|---|
| **OPRF** | 61.82 | 1.40 | 2.54 |
| **PosRF** | 62.31 | 1.21 | 2.19 |
| **SADERF** | 62.33 | 1.39 | 2.69 |

| Retrieval | acc | time | mem |
|---|---|---|---|
| **PosRF** | 77.32 | 1.22 | 2.01 |
| **SADERF** | 78.85 | 1.43 | 2.54 |
| **OPRF** | 79.11 | 1.39 | 2.41 |

