# OpenReview forum: "Spectraformer: A Unified Random Feature Framework for Transformer"
_ICLR.cc/2025/Conference — ICLR 2025 Conference Withdrawn Submission_

### Official Review · Reviewer_CtPi · 2024-10-26

**Soundness:** 2
**Presentation:** 2
**Contribution:** 2
**Rating:** 3
**Confidence:** 3

**Summary:**

Attention mechanism in Transformers have quadratic time complexity in the sequence length, which can create computational bottlenecks. Various methods have been proposed to address this, with kernel-based approaches for "linearizing" attention gaining significant interest in the ML community. In this work, the authors introduce Spectraformer, an unified framework for approximating and learning the kernel function in linear Transformers. The authors explore a range of sampling techniques and random features, demonstrating that different tasks benefit from different combinations of kernel methods, with no single method emerging as superior across all cases. The empirical results are evaluated using the LRA benchmark.

**Strengths:**

-  Good overview and discussion about various random feature mechanisms for linearizing attention in Transformers.
- Interesting observation that there is no clear random feature method excelling at all the tasks.

**Weaknesses:**

- Please rewrite Sec 3.4 in terms of pseudocode. Please do not point to specific lines of code.
- Writing needs to be improved, for example the concepts are mentioned before they are defined. Def 3.2 defines a valid component function even though this concept was mentioned in line 262-263.
- QMC is explored in the context of shift-invariant kernels in [1] and also in general random features  [2]. It feels incremental without any theoretical results as the authors are merely combining different matrices (that have been used to reduce variance) in some well-known modern random feature mechanism. It would be an extremely valuable work if the authors show that incorporating these matrices would lower variance over the orthogonal ones.
- The empirical results needs to be stronger. Specifically :
    * How does these methods perform in simply approximating the softmax kernel.
    * We generally see a quality gap when training Performers from scratch compared to regular transformers, does any of this combination close the gap? For ex. see performance gap between Performer-ViT vs Regular ViT (Fig 8. in [3]).
- As pointed out by the authors, different combinations work well for various tasks thus the paper should provide some guidance for practitioners on how to choose the right combination.

[1] Quasi-Monte Carlo Feature Maps for Shift-Invariant Kernels. Avron et al. JMLR 2016.

[2] Hybrid Random Features. Choromanski et al. ICLR 2022.

[3] From block-Toeplitz matrices to differential equations on graphs: towards a general theory for scalable masked Transformers. Choromanski et al. ICML 2022.

**Questions:**

In addition to the questions above, I have a few more additional questions.

- Why for any $\mathbf{W}$, $f$ = Trig a valid component function? (line 263)
- In line 284, what does it mean “among combination with trig RF”? This is true for positive random features as well (see Sec 4.2 in [1])
- Finally training the kernel to mimic the spikyness of softmax can lead to performance gains as observed in [2]. Does similar observations work in this setting?

[1] Rethinking Attention with Performers. Choromanski et al. ICLR 2021


[2] The Hedgehog & the Porcupine: Expressive Linear Attentions with Softmax Mimicry. Zhang et al. ICLR 2024

---

### Official Review · Reviewer_vnFR · 2024-10-28

**Soundness:** 2
**Presentation:** 3
**Contribution:** 2
**Rating:** 3
**Confidence:** 3

**Summary:**

The kernel approximation, based on Bochner’s theorem, enables the computation of linearized attention to be understood as a similarity computation using component functions $\phi(\cdot)$ and a learnable weight matrix $W$. Based on this interpretation, recent works have improved attention methods by presenting new component functions and parameterizations of the weight matrix.

This work further explores how various combinations of weight matrices and component functions, proposed in previous works, can improve the attention method, and refers to it as $Spectraformer$.

**Strengths:**

* This work presents the framework generalizing the random-feature based attention method.

**Weaknesses:**

* This work recombines the component functions $\phi(\cdot)$ and a learnable weight matrix $W$ presented in existing random-feature attention method, and does not present new idea for improving attention. Thus, the novelty of this work itself seems marginal.



* The benefits of exploring other combinations are not convincingly demonstrated. While it is possible that certain unexplored combinations of component functions and learnable weight matrices could improve accuracy, training time, or memory efficiency for some condition, this work does not clarify when or why certain combinations might be advantageous.

**Questions:**

* Based on Table 3, the performances seems significantly different depending on the combination of the component functions and a learnable weight matrix. Is there a guideline for selecting the best combination before training?





* When employing  $Spectraformer$,  it is unclear how to pick the better combinated component functions and weight matrix that can outperform the standard attention.

---

### Official Review · Reviewer_9vAS · 2024-10-31

**Soundness:** 3
**Presentation:** 3
**Contribution:** 2
**Rating:** 6
**Confidence:** 4

**Summary:**

The paper presents a family of Transformers, called Spectraformers, introducing a new class of linear low-rank attention mechanisms. In Spectraformer, the nonlinear map defining the transformation of queries/keys combines various nonlinear functions (applied to the queries/keys projected via Gaussian or learned projections). This formulation is general enough (Eq. 11) to cover as special cases various mechanisms introduced before for the unbiased estimation of the softmax kernel (including in particular celebrated positive random features). The Authors test presented mechanism on three tasks from the Long Range Arena benchmark.

**Strengths:**

The research on linear low-rank attention methods for Transformers is important for several practical reasons (fast inference, e.g. for on-device-deployment, etc.) and this paper aims to improve the existing methods in the field. The presented extension is sound and the idea to represent projections as learnable vectors rather than vectors sampled from a fixed probabilistic distribution is a neat idea.  The experimental section presents a comprehensive comparison with several related methods.

**Weaknesses:**

The conclusions are stated pretty vaguely, we read: " Our empirical findings indicate that different kernels are good at different tasks and that kernel choice is fundamental to performant models". This is not a particularly informative statement. Learning the projections rather than taking them from a fixed distribution might introduce additional computational costs. This should be discussed in depth in the paper. Finally, LRA is a pretty old benchmark for testing long-range-attention Transformers. The paper would benefit from applying more recent benchmarks.

**Questions:**

1. What is the extra computational and memory footprint corresponding to the additional parameters introduced in this mechanism ?
2. The most general formula presented in the paper (Eq. 11) does not provide an unbiased estimation of the softmax kernel. Is there a subset of the instantiations coming from Eq. 11 that is a strict superset of the previously known low-rank linear attention mechanisms providing unbiased estimation of the softmax kernel and still providing unbiasedness ?
3. The Authors say: " Our empirical findings indicate that different kernels are good at different tasks and that kernel choice is fundamental to performant models". More details would be important. What can we say about right kernels for particular applications (e.g. NLP, Vision, long-range-, short-range-attention tasks, etc.). Did the conducted experiments provide any additional insight to shed new light on that ?

---

### Official Review · Reviewer_hAqL · 2024-11-03

**Soundness:** 3
**Presentation:** 4
**Contribution:** 2
**Rating:** 3
**Confidence:** 4

**Summary:**

This paper investigates the linearization of attention mechanisms in Transformers using kernel approximation. The authors propose to unify the wide range of work on linearizing the attention kernel in a unified framework. To that end they propose to look at the linear kernel methods as approximating the weight or the component function. In the process they propose new linear methods that fill the gap. Their empirical analysis showcases that other combination methods do perform slightly better than previous SOTA methods.

**Strengths:**

-- The paper’s proposed unification exposes gaps which can be filled with novel combination of linear kernel methods

-- The paper is well written and easy to follow

-- On LRA benchmark the novel combinations show some promise

**Weaknesses:**

-- Novelty: The formulation is somewhat of a repeat of chowdhury et al 2022’s formulation. Even though it is more complete with more component functions and weights.

-- Empirical evaluation is insufficient. LRA dataset on its own is not sufficient to evaluate which combination works best. The LRA benchmark is old and doesn’t satisfy the the current requirements. IMHO for this paper to pass the acceptance threshold, I would want a much more thorough evaluation, on multiple benchmark dataset and multiple modalities. Then any conclusion made from the analysis would be useful for the practitioner.

-- it is unfortunate that there isn't a clear winner or new novel recommendation that came out of the emperical analysis (which the author would recommend to a practitioner)

**Questions:**

Please see the weakness section, my ratings for the paper is based on the weakness (especially the second point).

---

### Note · Authors · 2024-11-23

**Comment:**

We would like to withdraw our submission. We sincerely thank the reviewers and organizers for their time and effort.

**Withdrawal Confirmation:**

I have read and agree with the venue's withdrawal policy on behalf of myself and my co-authors.